# Drugging the PI3K/AKT/mTOR Pathway in ER+ Breast Cancer

**DOI:** 10.3390/ijms24054522

**Published:** 2023-02-24

**Authors:** Carla L. Alves, Henrik J. Ditzel

**Affiliations:** 1Department of Cancer and Inflammation Research, Institute of Molecular Medicine, University of Southern Denmark, 5000 Odense, Denmark; 2Department of Oncology, Institute of Clinical Research, Odense University Hospital, 5000 Odense, Denmark; 3Academy of Geriatric Cancer Research (AgeCare), Odense University Hospital, 5000 Odense, Denmark

**Keywords:** estrogen receptor-positive breast cancer, PI3K/AKT/mTOR pathway, CDK4/6 inhibitor

## Abstract

The frequent activation of the PI3K/AKT/mTOR pathway and its crucial role in estrogen receptor-positive (ER+) breast cancer tumorigenesis and drug resistance has made it a highly attractive therapeutic target in this breast cancer subtype. Consequently, the number of new inhibitors in clinical development targeting this pathway has drastically increased. Among these, the *PIK3CA* isoform-specific inhibitor alpelisib and the pan-AKT inhibitor capivasertib were recently approved in combination with the estrogen receptor degrader fulvestrant for the treatment of ER+ advanced breast cancer after progression on an aromatase inhibitor. Nevertheless, the clinical development of multiple inhibitors of the PI3K/AKT/mTOR pathway, in parallel with the incorporation of CDK4/6 inhibitors into the standard of care treatment in ER+ advanced breast cancer, has led to a multitude of available therapeutic agents and many possible combined strategies which complicate personalizing treatment. Here, we review the role of the PI3K/AKT/mTOR pathway in ER+ advanced breast cancer, highlighting the genomic contexts in which the various inhibitors of this pathway may have superior activity. We also discuss selected trials with agents targeting the PI3K/AKT/mTOR and related pathways as well as the rationale supporting the clinical development of triple combination therapy targeting ER, CDK4/6 and PI3K/AKT/mTOR in ER+ advanced breast cancer.

## 1. Introduction

The phosphatidylinositol 3-kinase (PI3K)/AKT/mammalian target of the rapamycin (mTOR) pathway is involved in various crucial cellular functions such as growth, proliferation, metabolism and survival [1,2]. Activation of this signaling pathway is triggered by receptor tyrosine kinases (RTK) or G protein-coupled receptors (GPCR) located at the plasma membrane, which induce the recruitment of class I PI3K protein by adaptor proteins, such as insulin receptor substrate (IRS). This leads to conversion of phosphatidylinositol 4,5-bisphosphate (PIP2) to phosphatidylinositol 3,4,5-trisphosphate (PIP3) (Figure 1). PIP3 functions as a second messenger that recruits and activates AKT, which phosphorylates and inactivates tuberous sclerosis complex (TSC) 1/2, a negative regulator of mTORC1. Ultimately, activation of mTORC1 induces S6- and 4E-BP1-mediated protein and lipid synthesis and decreased autophagy, resulting in cell growth and proliferation. Importantly, mTORC1 regulates a negative feedback loop that prevents overactivation of the pathway at AKT. The downstream effects of PI3K activation can be antagonized by the tumor suppressor phosphatase and tensin homolog (PTEN) through dephosphorylation of PIP3 back to PIP2 [2,3]. Activation of RTK and GPCR also induces RAS/RAF/MEK/ERK signaling, which further reinforces the activation of PI3K (Figure 1) [4,5,6].

Abnormal activation of the PI3K/AKT/mTOR pathway often promotes excessive cell growth and resistance to apoptosis and is commonly implicated in a wide variety of cancers [7]. Alterations of this pathway are particularly frequent in breast cancer, which remains the most common cancer and second cause of cancer death in women worldwide [8]. Indeed, approximately 70% of all breast tumors exhibit an alteration that renders the PI3K/AKT/mTOR pathway hyperactivated [9]. These often include hotspot single amino acid substitutions in the p110α subunit of the PI3K, encoded by *PIK3CA* [10,11]. Additionally, *AKT1-3* mutations and/or amplifications, *PDK1* amplification, *PTEN* and *TSC1/2* inactivating mutation, and deletion or epigenetic silencing, which cause hyperactivation of the pathway, have also been found in breast cancer and have been suggested to hold prognostic or predictive value [10,12,13,14]. The frequency of these alterations may vary across the different breast cancer subtypes. Estrogen receptor-positive (ER+) breast cancer represents the largest breast cancer subtype and is often associated with mutations of *PIK3CA* at substantially higher rates than triple-negative breast cancer (TNBC). Although mutations in individual genes occur rarely, combined *PIK3CA*-, *AKT1*- and *mTOR*-activating mutations together with inactivation/loss of *PTEN* are observed in approximately 25–30% of all TNBC [10,15,16,17].

## 2. The Role of the PI3K/AKT/mTOR Pathway in ER+ Breast Cancer

Interaction between the ER and PI3K/AKT/mTOR pathways occurs at multiple nodes of each pathway (Figure 2). Activation of the PI3K/AKT/mTOR pathway induces estrogen-independent ER transcriptional activity through phosphorylation of ERα by AKT or S6K1 [18,19]. Conversely, ER target gene expression activates upstream effectors of the PI3K/AKT/mTOR pathway, such as RTKs, receptor ligands and adaptors [20]. Furthermore, ER also activates the PI3K/AKT/mTOR pathway by direct binding to the p85α regulatory subunit of PI3K [21]. Approximately 30–40% of all ER+ breast tumors exhibit an activating mutation of *PIK3CA*, which can either increase the catalytic activity or cause the retention of the p110α subunit, thereby promoting excessive cell multiplication and resistance to apoptosis [22,23]. *PIK3CA*-mutated tumors have been associated with ligand-independent activation of ER, which causes poor response to antiestrogens compared to wild-type tumors [20,24]. Furthermore, activation of the PI3K/AKT/mTOR pathway has been demonstrated as a mechanism of resistance to long-term estrogen deprivation [25]. Indeed, endocrine-resistant preclinical models showed increased phosphorylation levels of PI3K and mTOR substrates, and targeted inhibition of these molecules impaired cell growth and improved response to endocrine therapy [26,27,28,29]. Conversely, compensatory ER transcriptional expression is observed following inhibition of the PI3K/AKT/mTOR signaling, and co-inhibition of ER and PI3K showed synergistic effects in ER+ *PIK3CA*-mutated preclinical models, supporting the coregulation of the two pathways [2,24]. Together, these data provided the rationale for clinical investigations of combined ER and PI3K/AKT/mTOR inhibition in endocrine therapy-resistant ER+ breast cancer.

## 3. Key Targetable Regulators of the PI3K/AKT/mTOR Pathway in ER+ Breast Cancer

The frequent activation of the PI3K/AKT/mTOR pathway observed in ER+ breast cancer and its implication in the development of acquired endocrine resistance has made it a key target for pharmacologic intervention in this patient population. Indeed, a wide range of agents targeting regulators of this pathway have been investigated in preclinical and clinical studies, including allosteric inhibitors of mTORC1, pan- or isoform-specific PI3K inhibitors, ATP-competitive inhibitors of mTORC1/mTORC2, dual PI3K/mTOR inhibitors and allosteric or catalytic inhibitors of AKT (Table 1). Despite such efforts, only a handful of these agents have been granted approval by the FDA/EMA for the treatment of ER+ advanced breast cancer, primarily due to dose-limiting toxicity and consequent use of subtherapeutic dosages that result in incomplete pathway inhibition. Furthermore, the disruption of negative feedback loops, such as the mTORC1/S6K1 negative loop at IRS1 (Figure 1), caused by PI3K/AKT/mTOR inhibitors, paradoxically triggers activation of the pathway.

### 3.1. mTOR Inhibitors

mTORC1 inhibitors, such as everolimus and temsirolimus, are allosteric irreversible inhibitors of mTORC1-dependent phosphorylation of S6K1 [2,30]. Results from the BOLERO-2 and TAMRAD clinical trials showed that the addition of everolimus to either exemestane or tamoxifen was associated with longer progression-free survival (PFS) compared to either exemestane or tamoxifen alone in ER+ advanced breast cancer patients who had progressed on an aromatase inhibitor (AI) (BOLERO-2, PFS 7.8 vs. 3.2 months, *p* < 0.0001; TAMRAD, PFS 8.6 vs. 4.5 months, *p* < 0.01) [31,43]. These findings led to FDA and EMA approval of combined everolimus and endocrine therapy for metastatic ER+ breast cancer after progression on an AI. More recently, results from the PrE0102 clinical trial showed that combined everolimus and the estrogen receptor downregulator, fulvestrant, improved PFS more compared to the fulvestrant alone (10.3 vs. 5.1 months, *p* = 0.02) in ER+ advanced breast cancer patients previously treated with an AI [32]. However, the severe toxicity of everolimus has limited its use in the clinic. Currently, several clinical trials are evaluating the efficacy of the dual mTORC1/2 inhibitors AZD2014 and sapanisertib, which produce a more complete blockade of mTORC by inhibiting both mTORC1-dependent phosphorylation of S6K1 and mTORC2-dependent phosphorylation of AKT, and show activity in mTORC1-mutated everolimus-resistant tumors [44,45,46]. Importantly, early clinical trials investigating mTORC1/2 inhibitors found higher single-agent activity than previously observed with mTORC1 inhibitors in various solid tumors, including ER+ breast cancer [47,48].

### 3.2. Pan-PI3K Inhibitors

Several pan-PI3K inhibitors have been developed, including buparlisib, pilaralisib and pictilisib, which block all isoforms of class IA PI3Ks [49]. These agents are associated with a high toxicity profile that precludes administration of an effective dose and does not significantly improve tumor growth inhibition compared to endocrine therapy alone. Results from the phase III BELLE-2 trial showed that combined buparlisib and fulvestrant modestly improved PFS compared to fulvestrant alone in ER+ advanced breast cancer patients who progressed on an AI (6.9 vs. 5.0 months, *p* < 0.001) [33]. Notably, a sub-analysis in this trial showed that there was a substantial improvement in PFS for patients with *PIK3CA* mutations treated with the combination compared to those treated with endocrine therapy alone (7.0 vs. 3.2 months, *p* < 0.001). More recently, results from the phase III BELLE-3 trial showed a modest, albeit statistically significant, improvement in PFS in the combined buparlisib- and fulvestrant-treated arm compared to fulvestrant alone in ER+ advanced breast cancer patients after progression on endocrine therapy and everolimus (3.9 vs. 1.8 months, *p* = 0.0003) [34]. These data, together with the high rates of serious adverse effects observed with these agents, limit further clinical development of buparlisib in this patient population [5]. In the FERGI clinical study, addition of pictilisib to fulvestrant did not significantly improve PFS in ER+ advanced breast cancer resistant to treatment with an AI in the adjuvant or metastatic setting [35].

### 3.3. Isoform-Specific PI3K Inhibitors

Several p110α isoform-specific inhibitors have been developed, including alpelisib and taselisib [49], that block the response of the PI3K/AKT/mTOR pathway to several growth stimuli [50]. The first isoform-specific inhibitor that was clinically investigated was alpelisib, which showed preferential activity in *PIK3CA*-mutated tumors [51,52]. The results from the large phase III SOLAR-1 trial showed improved PFS in the group receiving combined alpelisib and endocrine therapy compared to endocrine therapy alone in *PIK3CA*-mutated ER+ metastatic breast cancer patients previously treated with antiestrogen therapy. This led to the approval of this combination by the FDA and EMA [36]. In contrast, the NEO-ORB trial showed no improvement in the overall response rate (ORR) and pathologic complete response (pCR) with the addition of alpelisib to letrozole in the neoadjuvant setting of either *PIK3CA*-mutated or wild-type ER+ early breast cancer [37]. Notably, preliminary results of the phase II BYLIEVE trial that investigated combined alpelisib and endocrine therapy (letrozole or fulvestrant) in patients with *PIK3CA*-mutated ER+ advanced breast cancer after progression on combined CDK4/6 inhibitor and endocrine therapy showed a longer PFS for patients previously treated with CDK4/6 inhibitor and an AI, which supported the clinical relevance of alpelisib in this subpopulation [38]. Taselisib is another *PIK3CA*-mutated isoform-specific inhibitor that has been evaluated in the phase III SANDPIPER trial. A modest, albeit statistically significant, improvement in PFS for combined taselisib and fulvestrant compared to fulvestrant alone was observed in patients with ER+ advanced tumors who had progressed during or after AI treatment, irrespective of the *PIK3CA* mutation status (7.4 vs. 5.4 months, *p* = 0.0037) [39]. Additionally, the phase II LORELEI trial found no significant difference in pCR between combined taselisib and letrozole versus letrozole alone as neoadjuvant treatment in patients with stage I–III, operable, ER+/HER2− negative (HER2−) breast tumors with or without *PIK3CA* mutation [40]. Both SANDPIPER and LORELEI trials showed high rates of serious adverse effects that resulted in treatment discontinuation in 17% and 11%, respectively, of the taselisib-treated patients and precluded further clinical development of this drug.

### 3.4. Pan-AKT Inhibitors

Development of isoform-specific AKT inhibitors has been challenging due to the high structural similarity between the three isoforms (AKT1/2/3) [53]. Pan-AKT inhibitors include ATP-kinase activity inhibitors, such as capivasertib and ipatasertib, and allosteric inhibitors, such as MK-2206. In the phase II FAKTION trial, combined fulvestrant and capivasertib significantly prolonged PFS compared to fulvestrant alone (10.3 months vs. 4.8 months, *p* = 0.0018) in patients with ER+ locally advanced or metastatic breast cancer who had relapsed or progressed on an AI [41]. Furthermore, a phase I study evaluating capivasertib as monotherapy or in addition to fulvestrant in heavily pre-treated ER+ advanced breast cancer patients harboring the *AKT1* E17K mutation showed favorable activity and tolerability of capivasertib as a single agent and in the combination regimen, suggesting the potential clinical utility of capivasertib in this patient population [12]. Recently, positive results from the phase III CAPItello-291 trial evaluating capivasertib in combination with fulvestrant versus fulvestrant alone in patients with ER+ advanced breast cancer after progression on endocrine therapy, with or without an CDK4/6 inhibitor, showed that the addition of capivasertib to endocrine therapy significantly improved PFS in the overall patient population, independent of the AKT mutational status (7.2 vs. 3.6 months, *p* < 0.001) (7.3 vs. 3.1 months, *p* < 0.001) [42]. This trial is currently ongoing to investigate the effect of combined capivasertib and fulvestrant on overall survival (OS), but these encouraging findings will likely lead to FDA approval for ER+ advanced breast cancer patients who progressed on endocrine therapy with or without a CDK4/6 inhibitor.

### 3.5. Dual PI3K/mTOR Inhibitors

There has been an increasing interest in the clinical development of agents that provide dual inhibition of both PIK3CA and mTOR, and thus, achieve a more complete blockade by inhibiting multiple points of the PI3K/AKT/mTOR pathway and bypassing negative feedback loops associated with reduced clinical efficacy. Due to the structural similarities of PI3K and mTOR, these dual inhibitors can target the active sites of both kinases. This leads to blockage both upstream and downstream of AKT, thus avoiding the problem of AKT activation following inhibition of the mTORC1–S6K–IRS1 negative feedback loop, which has been reported with mTOC1 blockers [54]. Therefore, dual PI3K/mTOR have been associated with higher anti-tumor activity, but, unfortunately, also a higher toxicity profile [55]. Dactolisib, voxtalisib, bimiralisib and gedatolisib are some of the agents that have been evaluated in phase I/II trials [5]. Notably, gedatolisib has recently received breakthrough therapy designation by the FDA to accelerate the development and regulatory review of this agent based on data from a Phase 1b trial that assessed the safety, tolerability and clinical activity of gedatolisib in combination with endocrine therapy and CDK4/6 inhibitor in ER+ advanced breast cancer that progressed on CDK4/6 therapy and an AI [56]. Consequently, gedatolisib is being evaluated in the phase III trial VIKTORIA-1 combined with fulvestrant with or without the CDK4/6 inhibitor palbociclib in this patient population (NCT05501886).

## 4. Determining the Optimal Point of Inhibition of the PI3K/AKT/mTOR Pathway in ER+ Breast Cancer

The recent explosion in the number and diversity of PI3K/AKT/mTOR inhibitors in clinical development creates the need for a rational approach to identify the tumor/patient that will benefit the most from a specific inhibitor. Thus, it is critical to identify the genomic contexts in which these various types of inhibitors show superior activity [57]. It is noteworthy that the successful introduction of these agents into the clinic is dependent on finding tolerable dosages that efficiently inhibit the pathway and achieve anti-tumor activity. Indeed, the development of compounds targeting the PI3K/AKT/mTOR pathway has been significantly precluded by the broad range of off- and on-target effects and associated toxicity, which most often include hyperglycemia, dermatitis and rashes, stomatitis, diarrhea and nausea and fatigue [58,59].

Based on early-phase clinical evidence, pan-PI3K and dual PI3K/mTOR inhibitors, by inhibiting all four isoforms of PI3K, may be better suited for tumors associated with multiple and heterogeneous molecular alterations in the PI3K/AKT/mTOR pathway [57]. Indeed, studies have shown responses to pan-PI3K inhibitors in both *PIK3CA*-mutated and wild-type tumors, which may exhibit pathway activation driven by molecular alterations in other pathway components [60,61]. Although pan-PI3K and dual PI3K/mTOR inhibitors are associated with similar adverse effects, more frequently reported with the latter, it is likely that the pan-PI3K inhibitors, due to their narrower activity profile and wider therapeutic window, are more suitable than dual PI3K/mTOR inhibitors for combined therapies with other targeted or cytotoxic agents [62,63,64]. Nevertheless, tumors exhibiting alterations downstream of PI3K but upstream of mTOR, such as the loss of *PTEN* or *TSC1/2*, may be more efficiently suppressed by dual PI3K/mTOR inhibitors, which target the pathway at multiple sites and thus show the broadest activity profile [65].

In contrast to pan-PI3K and dual PI3K/mTOR inhibitors, isoform-specific PI3K inhibitors exhibit a narrower activity, which make them more amenable to combination with other pathway inhibitors and may offer greater opportunities for optimized dosing and schedule of therapy [57]. Furthermore, the high selectivity of isoform-specific PI3K inhibitors implies that these agents may show higher activity in tumors with specific mutations, but a reduced efficacy in tumors with multiple PI3K/AKT/mTOR pathway alterations, and thus require biomarker-based patient selection [66,67].

AKT inhibitors may be particularly valuable in tumors with *PTEN* loss, which do not benefit from either pan- or isoform-specific PI3K-targeted agents [68,69,70]. Although PTEN alterations appear to be a strong indicator of AKT inhibitor efficacy, *PIK3CA*-mutant tumors may also benefit from AKT inhibition. The rationale for targeting *PIK3CA*-mutant tumors with AKT inhibitors lies in the AKT function of funneling all PI3K signaling activity [69,71]. Indeed, we and others have shown that sensitivity to AKT inhibitors was observed in both *PIK3C*A-mutant/*PTEN*-wild-type and *PIK3CA*-wild-type/*PTEN*-null breast cancer cell lines, and sensitivity to AKT inhibitors correlated with SGK and p-AKT expression levels [72,73,74,75]. Concordantly, the FAKTION trial showed that *PIK3CA*-mutant ER+ metastatic breast tumors benefit from combined capivasertib and fulvestrant [41]. In ER+ breast tumors with *AKT1* E17K gain-of-function mutations, which promote constitutive activation of the downstream pathway, treatment with the AKT capivasertib yielded tumor regression [76]. Notably, additional genetic alterations in the PI3K/AKT/mTOR pathway were associated with prolonged PFS, suggesting that multiple alterations within the pathway might further sensitize *AKT1*-E17K-mutated tumors to AKT inhibitors [71]. In spite of this, it has been demonstrated that PI3K controls additional parallel, independent, oncogenic pathways, such as ERK signaling, which AKT inhibitors may fail to block [77]. Thus, not all *PIK3CA*-mutated breast cancer models might benefit from AKT blockers, as some may promote cell growth through an AKT-independent axis, such as PDK1/SGK3/mTORC1 [73,78]. Notably, results from recent clinical studies supported the addition of an AKT inhibitor to first-line paclitaxel treatment of TNBC with alterations in the *PI3KCA*/*AKT1*/*PTEN* axis [79,80]. These findings have not yet been demonstrated with other inhibitors of the PI3K/AKT/mTOR pathway [81,82].

Regarding mTORC inhibitors, dual mTORC1/2 inhibitors have demonstrated greater efficacy than mTORC1 inhibitors in clinical trials, likely due to their ability to inhibit both mTORC complexes, thus bypassing the activation of AKT by mTORC2. Furthermore, dual mTORC1/2 inhibitors function as catalytic inhibitors, in contrast to the allosteric mTORC1 inhibitors, which may explain the greater inhibitory activity against mTORC1 of some dual mTORC1/2 inhibitors compared to mTORC1 inhibitors [44]. Importantly, the profound inhibition of 4E-BP1 achieved via dual mTORC1/2 inhibition, but not with mTORC1 inhibitors, may explain the differential anti-tumor effect between these two classes of drugs [47,83,84]. However, some mTORC1/2 agents have only been capable of causing transient tumor growth inhibition, comparable to mTORC1 inhibitors, which may suggest similar mechanisms of resistance between these agents [85].

## 5. Combinatorial and Sequential Treatments with PI3K/AKT/mTOR Inhibitors in ER+ Breast Cancer

Although the genomic landscape of breast cancer supports development of therapeutic strategies targeting the PI3K/AKT/mTOR axis, complete blockage of this pathway remains elusive. A common limitation of all inhibitors of the PI3K/AKT/mTOR pathway is the compensatory activation of multiple upstream tyrosine kinase receptors and other compensatory mechanisms that can reactivate the PI3K pathway and impair the efficacy of these agents [71]. Therefore, optimization of drug combination regimens and biomarker-based population refinement is urgently warranted to improve clinical responses to PI3K/AKT/mTOR inhibitors in ER+ breast cancer [24,71,86].

Notably, the last few years have seen impressive improvement in the clinical outcomes of ER+ advanced breast cancer patients as a result of the incorporation of CDK4/6 and PI3K inhibitors in the standard of care treatment. Although the isoform-specific PI3K inhibitor alpelisib has proven to significantly improve PFS in *PIK3CA*-mutated tumors in combination with fulvestrant, this agent is associated with serious grade 3–4 adverse effects [36]. In contrast, CDK4/6 inhibitors showed improved PFS and OS compared to endocrine therapy alone in both endocrine-sensitive or endocrine-resistant tumors, with acceptable toxicity profiles, which can be successfully managed using drug dose reduction or withdrawal [87,88,89,90,91,92,93]. However, the development of resistance to CDK4/6 inhibitors is inevitable, and one of the suggested resistance mechanisms is the convergence of the cell-cycle and PI3K/AKT/mTOR pathways. Indeed, we and others have shown that upregulation of p-AKT, PDK1, p70S6K and loss of *PTEN* are associated with resistance to CDK4/6 inhibitors preclinically, and treatment with agents targeting the PI3K/AKT/mTOR pathway can overcome CDK4/6 inhibitor resistance [75,94,95,96].

Furthermore, the crosstalk between the ER, cyclin D-CDK4/6 and PI3K/AKT/mTOR pathways has been demonstrated extensively in preclinical studies, with cyclin D1 acting as a common node (Figure 3) [97]. Binding of cyclin D1 to CDK4/6 induces Rb phosphorylation and subsequent uncoupling from E2F, which promotes G1-S phase cell cycle progression [98]. Importantly, estrogen promotes cyclin D1 transcription and, conversely, cyclin D1 and S6K can cause ligand-independent ER transcription. Furthermore, AKT-mediated inhibition of GSK3β stabilizes cyclin D1 from proteolytic degradation [99]. Inhibition of PI3K results in enhanced ER transcriptional activity, which can be overcome, at least in part, by inhibition of CDK4/6. Conversely, treatment with an CDK4/6 inhibitor causes incomplete cell cycle arrest that can be more efficiently blocked by the addition of PI3K inhibition [95,100]. Together, the convergent effects and the complex intersection of these three interrelated pathways supported the recent clinical development of triple therapies targeting PI3K/AKT/mTOR, CDK4/6 and ER to further improve the clinical outcome in ER+ advanced breast cancer (Table 2) [97]. Most of these trials are currently enrolling patients or have just completed patient recruitment, and preliminary data have not yet been reported. Recently, a phase Ib trial testing the triple combination of CDK4/6 inhibitor palbociclib, *PIK3CA*-isoform-specific inhibitor taselisib and fulvestrant showed tolerability at pharmacodynamically-active doses and promising efficacy in heavily *PIK3CA*-mutated ER+/HER2− advanced breast cancer [101]. Disappointingly, another trial demonstrated high toxicity with a triple combination of either *PIK3CA*-isoform-specific inhibitor alpelisib or pan-PI3K inhibitor buparlisib with CDK4/6 inhibitor ribociclib and fulvestrant [97]. Nevertheless, numerous ongoing clinical trials are continuing to evaluate alpelisib in different triple combinations in ER+ advanced breast cancer subpopulations, and these results are crucial for definitive conclusions.

Critical questions remain to be answered: First, whether patients with *PIK3CA*-mutated ER+ breast cancer should receive alpelisib or CDK4/6 inhibitor plus endocrine therapy as first-line therapy in the advanced setting. Based on the final results from the MONALEESA-2/3 (CDK4/6 inhibitor ribociclib plus fulvestrant/letrozole) and SOLAR-1 (PI3K inhibitor alpelisib plus fulvestrant) trials, combined ribociclib and endocrine therapy was associated with statistically significantly longer PFS and OS, whereas combined alpelisib and fulvestrant showed a significant improvement in PFS, but the prolongation of OS did not reach statistical significance [36]. These data might favor the selection of ribociclib over alpelisib as the choice for endocrine therapy in the first-line setting, but this leads to another critical question: In order to overcome resistance to the CDK4/6 inhibitor, should patients receive upfront triple therapy with PI3K, CDK4/6 and ER blockers, or standard double combinations with ribociclib or alpelisib and, upon progression, switch to triple combination or an alternative double combination? Preclinical studies have shown that upfront triple combination with endocrine therapy, CDK4/6 and PI3K or mTOR inhibitors achieved greater cell cycle arrest, induced apoptosis and tumor regression in in vitro and in vivo CDK4/6 inhibitor-naïve models of advanced ER+ breast cancer, but not in models of acquired resistance to the CDK4/6 inhibitor [94,95]. In contrast, preliminary data from the phase II BYLieve trial, which investigates the efficacy of combined alpelisib and fulvestrant, showed that this treatment is also effective in *PIK3CA*-mutated tumors previously treated with the CDK4/6 inhibitor [38]. In line with this, we have recently shown that triple combination with fulvestrant, palbociclib/ribociclib and AKT inhibitor capivasertib or isoform-specific PI3K inhibitor alpelisib efficiently suppressed tumor growth in in vitro and in vivo models of resistance to combined endocrine therapy and the CDK4/6 inhibitor [75,102]. Furthermore, we showed that switching the CDK4/6 inhibitor for the AKT/PI3K inhibitor in combination with the endocrine therapy backbone did not prevent tumor outgrowth in combined endocrine- and CDK4/6 inhibitor-resistant models [75,102]. Importantly, our study showed that a double combination with fulvestrant plus either the AKT/PI3K or CDK4/6 inhibitor efficiently blocked the growth of endocrine and CDK4/6 inhibitor-sensitive cells, which exhibit lower levels of phospho-AKT compared to resistant cells [75,102]. These findings highlight the urgent need for biomarkers for patient selection to optimize PI3K/AKT/mTOR-targeted therapy in ER+ advanced breast cancer. Matured data from ongoing multi-armed randomized clinical trials incorporating biomarker-based patient stratification will be crucial to fully answer these questions.

## 6. Conclusions

The treatment landscape for ER+ advanced breast cancer has improved significantly with the addition of the CDK4/6 inhibitors to endocrine therapy as standard treatment. However, all patients will eventually progress on this combined therapy and new rational therapeutic strategies are required upon progression. The crucial role of the PI3K/AKT/mTOR pathway in ER+ breast cancer tumorigenesis and treatment response has been demonstrated in numerous preclinical and clinical studies. Incorporation of a PI3K/AKT/mTOR inhibitor functions synergistically with endocrine therapy and CDK4/6 inhibitors to inhibit tumor growth and prevent or overcome resistance to standard therapy in ER+ metastatic breast cancer. This has led to a dramatic increase in the number of clinical studies investigating drugs targeting this pathway, and to the approval of the mTOR inhibitor everolimus and the PIK3CA inhibitor alpelisib combined with endocrine therapy in ER+ advanced breast cancer patients previously treated with antiestrogen therapy. Although multiple regimens have been suggested in different lines of therapy, the optimal treatment sequencing and combinatorial strategy in this clinical setting remain undefined. Currently, the CDK4/6 inhibitor continues to be the preferred choice for first-line combined treatment with endocrine therapy due to its better toxicity profile compared to either alpelisib or everolimus. Upon progression on combined CDK4/6 inhibitor and endocrine therapy, subsequent combined therapy will frequently switch to a PI3K/AKT/mTOR inhibitor with endocrine therapy. In ***PIK3CA***-mutated ***PTEN*** wild-type tumors, alpelisib or an alternative α-specific PI3K inhibitor will likely remain the first choice. For tumors with multiple alterations in the pathway, particularly ***PTEN*** loss and ***AKT*** mutations, dual PI3K-mTOR or pan-AKT inhibitors are preferred, with the latter showing the advantage of lower toxicity and better suitability to combination therapy. Nevertheless, preclinical data suggest that switching endocrine therapy partners from the CDK4/6 to PI3K/AKT/mTOR inhibitor will not be sufficient to efficiently overcome drug resistance, and tougher therapeutic strategies may be required. Indeed, upfront triple-targeted therapy might be needed in tumors pretreated with endocrine therapy, with or without the CDK4/6 inhibitor, whereas treatment-naive tumors may significantly benefit from standard double combinations. Results from ongoing randomized clinical trials investigating the optimal sequencing and combinations including these targeted agents with biomarker-based population refinement are warranted to fully optimize combinatorial strategies targeting PI3K/AKT/mTOR, CDK4/6 and ER in ER+ advanced breast cancer.

## Figures and Tables

**Figure 1 ijms-24-04522-f001:**
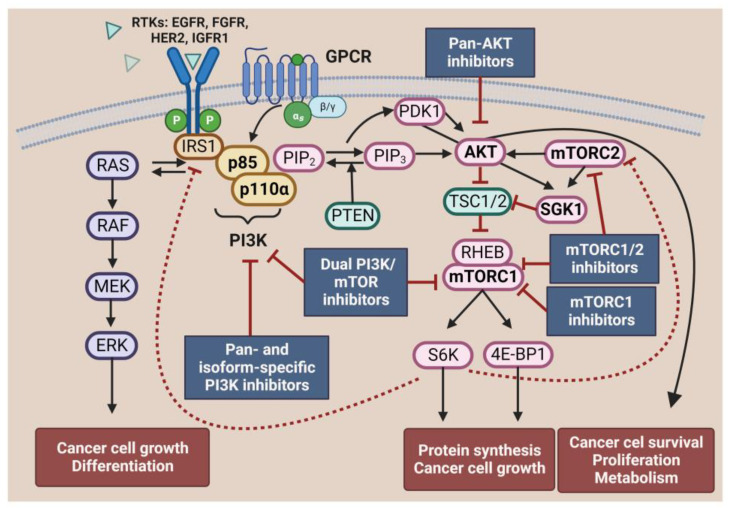
Schematic representation of the PI3K/AKT/mTOR pathway and its targetable regulators. RTK activation stimulates PI3K to convert PIP2 to PIP3, which recruits PDK1, AKT and mTORC2 to the plasma membrane. Both PDK1 and mTORC2 activate AKT, which activates mTORC1. A negative feedback loop is induced by mTORC1 at IRS1. The negative regulator PTEN converts PIP3 back to PIP2. GPCR, G protein-coupled receptor; RTK, receptor tyrosine kinase; IRS1, insulin receptor substrate 1; PDK1, phosphoinositide-dependent kinase 1; PIP2, phosphatidylinositol 4,5-bisphosphate; PIP3, phosphatidylinositol 3,4,5-trisphosphate.

**Figure 2 ijms-24-04522-f002:**
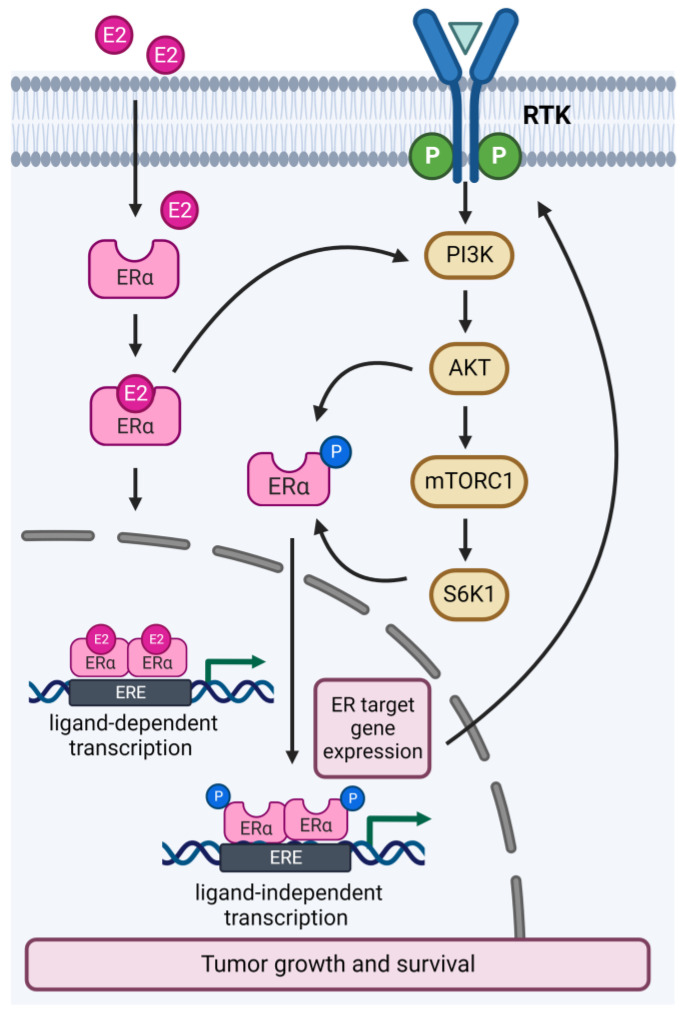
Crosstalk between the PI3K/AKT/mTOR and estrogen receptor pathways in ER+ breast cancer. The ER and PI3K/AKT/mTOR pathways interact directly and indirectly at multiple nodes of each pathway. E2, estrogen; ER, estrogen receptor; ERE, estrogen response elements; RTK, receptor tyrosine kinase.

**Figure 3 ijms-24-04522-f003:**
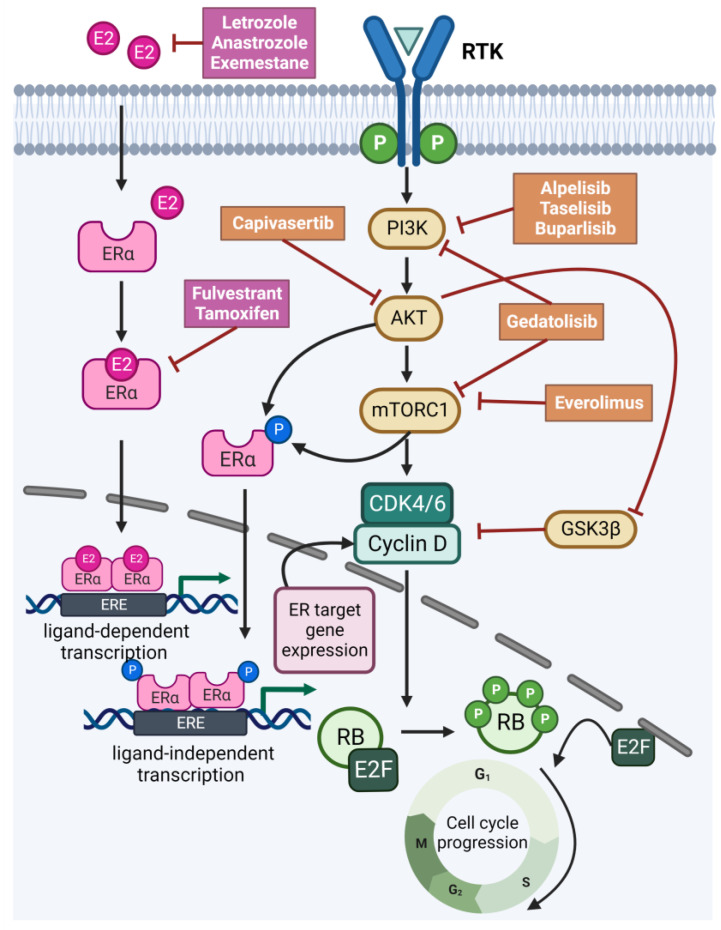
The convergence of the PI3K/AKT/mTOR, cyclin D-CDK4/6-RB and estrogen receptor pathways in ER+ breast cancer. Cyclin D1 functions as a common node by regulating the cell cycle through binding to CDK4/6, inducing ligand-independent activation of ER that can conversely induce cyclin D1 expression, which is further upregulated and stabilized by downstream effectors of the PI3K/AKT/mTOR pathway. Drugs targeting various regulators of the three pathways are also depicted. E2, estrogen; ER, estrogen receptor; ERE, estrogen response elements; RTK, receptor tyrosine kinase.

**Table 1 ijms-24-04522-t001:** Selected clinical trials with PI3K/AKT/mTOR inhibitors in ER+/HER2− advanced breast cancer.

Target	Drug	Clinical Trial (Phase)	Patient Population	Regimen	Outcome	FDA/EMA Approval	Reference
mTORC1	Everolimus	BOLERO-2 (III)	ER+/HER2− mBC after AI, *n* = 724	Exemestane ± everolimus	mPFS 6.9 vs. 2.8 months, HR 0.38, *p* < 0.001	Yes	[30]
		TAMRAD (II)	ER+/HER2− mBC after AI, *n* = 111	Tamoxifen ± everolimus	6-month CBR 61% vs. 42%; TTP 8.6 vs. 4.5 months	No	[31]
		PrE0102 (II)	ER+/HER2− mBC after AI, *n* = 131	Fulvestrant ± everolimus	mPFS 10.3 vs. 5.1, HR 0.61, *p* = 0.02	No	[32]
Pan-PI3K	Buparlisib	BELLE-2 (III)	ER+/HER2− locally aBC or mBC after AI, *n* = 1147	Fulvestrant ± buparlisib	mPFS 6.9 vs. 5.0 months, *p* < 0.001; *PIK3CA-*mut mPFS 7 vs. 3.2 months, HR 0.56, *p* < 0.001	No	[33]
		BELLE-3 (II)	ER+/HER2− locally aBC or mBC after ET + everolimus *n* = 432	Fulvestrant ± buparlisib	mPFS 3.9 vs. 1.8 months, HR 0.67, *p* = 000030	No	[34]
	Pictilisib	FERGI (II)	ER+/HER2− mBC Al-resistant *n* = 229	Fulvestrant ± pictilisib	mPFS 6.6 vs. 5.1 months, HR 0.74, *p* = 0.096	No	[35]
Isoform-specific PI3K	Alpelisib	SOLAR-1 (III)	ER+/HER2− mBC after ET, *n* = 572	Fulvestrant ± alpelisib	*PIK3CA*-mut mPFS 11.0 vs. 5.7 months, HR 0.65, *p* < 0.001	Yes	[36]
		NEO-ORB (II)	ER+/HER2− localized BC neoadjuvant *n* = 257	Letrozole ± alpelisib	ORR *PIK3CA*-mutant, 43% vs. 45%, *p* = 0.435, *PIK3CA*-wt, 63% vs. 61%, *p* = 0.611. No significant differences in pCR.	No	[37]
		BYLIEVE (II)	*PIK3CA*-mut ER+/HER2− mBC after CDK4/6i *n* = 127 (cohort A)	ET ± alpelisib	Median follow-up 11.7 months; pts without disease progression at 6 months: 50.4%	No	[38]
	Taselisib	SANDPIPER (III)	*PIK3CA*-mut ER+/HER2− locally aBC or mBC after AI *n* = 516	Fulvestrant ± taselisib	mPFS 7.4 vs. 5.4 months, HR 0.70, *p* = 0.0037	No	[39]
		LORELEI (II)	ER+/HER2− localized BC neoadjuvant *n* = 334	Letrozole ± taselisib	OR 39% vs. 50%; OR 1.55; *p* = 0.049; *PIK3CA*-mut OR 38% vs. 56%; OR 2.03, *p* = 0.033. No significant differences in pCR.	No	[40]
Pan-AKT	capivasertib	FAKTION (II)	ER+/HER2 locally advanced or mBC after AI *n* = 140	Fulvestrant ± capivasertib	mPFS 10.3 vs. 4.8 months, HR 0.58, *p* = 0.0044; mOS 29.3 vs. 23.4 months, HR 0.66, *p* = 0.035	No	[41]
		CAPItello-291 (III)	ER+/HER2− locally advanced or mBC, after ET ± CDK4/6i *n* = 708	Fulvestrant ± capivasertib	mPFS 7.2 vs. 3.6 months, *p* < 0.001; AKT-altered mPFS 7.3 vs. 3.1 months, *p* < 0.001	No	[42]

aBC, advanced breast cancer; AI, aromatase inhibitors; CDK4/6i, CDK4/6 inhibitors; CBR, clinical benefit rate; ET, endocrine therapy; HR, hazard ratio; mBC, metastatic breast cancer; mPFS, median progression-free survival; mOS, median overall survival; ORR, overall response rate; pCR, pathologic complete response; wt, wild-type.

**Table 2 ijms-24-04522-t002:** Clinical trials testing triple and sequential double combinations with inhibitors of the PI3K/AKT/mTOR, cyclin D/CDK4/6-RB and ER pathways in ER+ advanced breast cancer.

Target	Drugs and Regimen	Clinical Trial (Phase)	Patient Population (Actual or Estimated)	Outcome
mTORC1	everolimus (mTORi) + ribociclib (CDK4/6i) + exemestane (AI)	NCT02732119/TRINITI-1 (I/II)	ER+/HER2− mBC after CDK4/6i *n* = 104	CBR at week 24: 41.1%
	everolimus (mTORi) + palbociclib (CDK4/6i) + exemestane (AI)	NCT02871791 (I/II)	ER+/HER2− mBC *n* = 41	CBR at week 24: 18.8%
PI3K	taselisib (isPI3Ki)/pictilisib (pPI3Ki) + Ppalbociclib (CDK4/6i) + fulvestrant (SERD)	NCT02389842/PIPA (Ib)	ER+/HER2− mBC *n* = 25	ORR 37.5% CBR 58.3% mPFS 7.2 months
	fulvestrant (SERD) + ribociclib (CDK4/6i) ± alpelisib (isPI3Ki) or buparlisib (pPI3Ki)	NCT02088684 (I)	ER+/HER2− mBC *n* = 70	mPFS 7.2/11.0 vs. 7.2/11.0
	fulvestrant (SERD) + alpelisib (isPI3Ki) or ribociclib (CDK4/6i)	NCT05625087/SAFIR 03 (II)	ER+/HER2− mBC *PIK3CA*-mutated *n* = 162	NA
	letrozole (AI) + alpelisib (isPI3Ki) or ribociclib (CDK4/6i) ± ribociclib (CDK4/6i) or alpelisib (isPI3Ki)	NCT01872260 (Ib/II)	ER+/HER2− locally advanced or mBC *n* = 255	NA
	First-line letrozole (AI) + ribociclib (CDK4/6i) Second-line fulvestrant (SERD) + alpelisib (isPI3K)	NCT03439046/BioItaLEE (III)	ER+/HER2− mBC *n* = 287	NA
	inavolisib (isPI3Ki) + letrozole (AI) or fulvestrant (SERD) ± palbociclib (CDK4/6i)	NCT03006172 (I)	ER+/HER2− locally advanced or mBC, *PIK3CA*-mutated *n* = 256	NA
	OP-1250 (CERAN) + alpelisib (isPI3Ki) or ribociclib (CDK4/6i)	NCT05508906 (Ib)	ER+/HER2− mBC	NA
	CYH33 (isPI3Ki) + fulvestrant (SERD) or letrozole (AI) ± palbociclib (CDK4/6i)	NCT04856371 (Ib)	ER+/HER2− mBC *PIK3CA*-mut *n* = 228	NA
AKT	fulvestrant (SERD) + palbociclib (CDK4/6i) ± capivasertib (AKTi)	NCT04862663/CApitello-292 (Ib/III)	ER+/HER2− locally advanced or mBC after ET *n* = 700	NA
	fulvestrant (SERD) + palbociclib (CDK4/6i) ± ipatasertib (AKTi)	NCT04920708/FAIM (II)	ER+/HER2− mBC with/without ctDNA suppression *n* = 324	NA
PI3K/mTOR	fulvestrant (SERD) + palbociclib (CDK4/6i) ± gedatolisib (dual PI3K/mTORi) or alpelisib (isPI3Ki)	NCT05501886/VIKTORIA-1 (III)	ER+/HER2− locally advanced or mBC after CDK4/6i and AI with/without *PIK3CA* mutation *n* = 701	NA
	AZD2014 (dual PI3K/mTORi) + fulvestrant (SERD) + palbociclib (CDK4/6i)	NCT02599714/PASTOR (I)	ER+/HER2− locally advanced or mBC	NA

AI, aromatase inhibitors; CBR, clinical benefit rate; CDK4/6i, CDK4/6 inhibitor; CERAN, complete estrogen receptor antagonist; ET, endocrine therapy; isPI3Ki, isoform-specific PI3K inhibitor; mBC, metastatic breast cancer; mPFS, median progression-free survival; NA, not available; ORR, objective response rate; pPI3Ki, pan-PI3K inhibitor; SERD, selective estrogen receptor downregulator.

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
