# Peer review of "Drugging the PI3K/AKT/mTOR Pathway in ER+ Breast Cancer"

_ijms, 2023, doi:10.3390/ijms24054522_

Round 1

Reviewer 1 Report

The manuscript entitled "Drugging the PI3K/AKT/mTOR pathway in ER+ breast cancer: 2 What is the most effective therapeutic strategy? by Alves et al is a good and comprehensive study of the latest developments in this signaling pathway. However, some improvements need to be before its acceptance in IJMS.

  1. The last question in the title seems to be a mismatch for the review article. Consider modifying it.
  2. Can the authors talk a bit about the ER-negative breast cancers and their association with the PI3K/AKT/mTOR pathway? Because that is the main challenge in breast cancer treatment.
  3. I would like to hear more in the conclusions part of the review. The Conclusions part is small and does not do justice to the important info provided in the review. Write it in a way that it becomes a unified voice of the whole review article.

Author Response

Response to comments by Reviewer #1

The manuscript entitled "Drugging the PI3K/AKT/mTOR pathway in ER+ breast cancer: What is the most effective therapeutic strategy? by Alves et al is a good and comprehensive study of the latest developments in this signaling pathway. However, some improvements need to be before its acceptance in IJMS.

  1. The last question in the title seems to be a mismatch for the review article. Consider modifying it.

Our response: As suggested by the reviewer, we have modified the title to “Drugging the PI3K/AKT/mTOR pathway in ER+ breast cancer.“

  1. Can the authors talk a bit about the ER-negative breast cancers and their association with the PI3K/AKT/mTOR pathway? Because that is the main challenge in breast cancer treatment.

Our response: As requested by the reviewer, we have included information on the role of the PI3K/AKT/mTOR pathway in the triple-negative breast cancer (TNBC) subtype, including the most common alterations in this pathway and the promising findings from clinical trials investigating AKT inhibitors in TNBC (pages 2 and 10, respectively).  

  1. I would like to hear more in the conclusions part of the review. The Conclusions part is small and does not do justice to the important info provided in the review. Write it in a way that it becomes a unified voice of the whole review article.

Our response: As requested by the reviewer, we have expanded the Conclusion section (page 14).

Reviewer 2 Report

Dear authors, The topic you choose is of interest , but the article needs to be improved, taking into account the scientific level of this journal. Therefore I suggest the following:   1. line 28 full name for PI3K/AKT/mTOR 2. Line 30 abbreviation for G-protein coupled receptors 3. line 35 full name for TSC1/2 4. Line 40 full name for PTEN 5. In the introduction section, please write about  ER poz incidence 6. I don't agree with figure 1. In figure 1 appears that the growth factor? activates RTKs. What growth factor? RTKs are activated by a lot of factors, such as EGFR, hormones, and cytokines. Also,  I don't agree with the fact that PTEN inhibits PIP3. PTEN catalysis the dephosphorylation of PIP3 to PIP2.  According to the figure, looks like PTEN blocks PIP3 formation. Please modify this.  I suggest that the figure should include all the factors (including hormones) that activate RTKs and also   G-protein coupled receptors.  Maybe it would be better to appear information in the article about Ras-RAF ...ERK.  7. Maybe a figure including PI3K/AKT/mTOR and ER+ breast cancer development it would have a great impact on the readers 8. The conclusion section isn't very relevant.  9. They are a small number of references for a review article, regarding the topic, breast cancer.   Best regards,

Author Response

Response to comments by Reviewer #2

Dear authors, The topic you choose is of interest, but the article needs to be improved, taking into account the scientific level of this journal. Therefore I suggest the following:  

  1. line 28 full name for PI3K/AKT/mTOR; 2. Line 30 abbreviation for G-protein coupled receptors; 3. line 35 full name for TSC1/2; 4. Line 40 full name for PTEN

Our response: As requested by the reviewer, we have included the full names/abbreviations (page 1).

  1. In the introduction section, please write about ER poz incidence

Our response: As requested by the reviewer, we have included information on ER+ breast cancer incidence in the introduction (page 2).

  1. I don't agree with figure 1. In figure 1 appears that the growth factor? activates RTKs. What growth factor? RTKs are activated by a lot of factors, such as EGFR, hormones, and cytokines. Also, I don't agree with the fact that PTEN inhibits PIP3. PTEN catalysis the dephosphorylation of PIP3 to PIP2. According to the figure, looks like PTEN blocks PIP3 formation. Please modify this.  I suggest that the figure should include all the factors (including hormones) that activate RTKs and also   G-protein coupled receptors.  Maybe it would be better to appear information in the article about Ras-RAF ...ERK. 

Our response: We have modified Figure 1 according to the reviewer’s suggestions. We have also included information about the RAS/RAF/MEK/ERK pathway in the Introduction (page 1). 

  1. Maybe a figure including PI3K/AKT/mTOR and ER+ breast cancer development it would have a great impact on the readers

Our response: As suggested by the reviewer, we have included a new figure to show the role of the PI3K/AKT/mTOR pathway in ER+ breast cancer (new Figure 2) and described it in the text (page 3). We have also modified previous Figure 2 (new Figure 3) based on the modifications in new Figure 2. 

  1. The conclusion section isn't very relevant.

Our response: As requested by the reviewer, we have modified the Conclusion section (page 14).

  1. They are a small number of references for a review article, regarding the topic, breast cancer.

Our response: Following the modifications suggested by the reviewers, the manuscript now includes over 100 references, which we believe is an acceptable number for a review of this length.